# 4D printed deformation labels with machine learning for monitoring and preservation of respiring climacteric fruits

Xiuxiu Teng[1,2], Min Zhang [1,3] ✉, Arun S. Mujumdar[4] & Chunli Li[1]

4D printed labels that change color and shape were developed to achieve the dual functions of quality assessment and maintenance of respiring climacteric fruits. The effect of addition of essential oil emulsion and different geometric structures on deformation as well as the corresponding mechanisms were explored. Cast, 3D printed, and 4D printed labels were compared based on their responses to fruit quality and compatibility with machine learning. The addition of emulsion significantly affected the degree of deformation by altering the printing fidelity, hydrophilicity, and flexibility of the network structure. Geometric designs (Including printing layers, filament intersection angles, and infill ratios) changed both the direction and degree of deformation. Unlike cast and 3D printed labels, 4D printed labels simultaneously changed color and shape in response to variations in humidity and carbon dioxide levels in the package, enabling more accurate visual monitoring the turning points of fruit quality. The MobileNet model achieved recognition accuracy of 97% for 4D printed labels, which played an active role in achieving intelligent warnings. Additionally, deformation along with microstructure destruction positively impacted the controlled release of essential oils through a non-Fickian mechanism, resulting in a better preservation effect.

Approximately $60 million worth of fruits and vegetables are wasted annually, wherein respiring climacteric fruits are often not consumed within their optimal consumption time[1]. This delay in consumption often leads to spoilage as a result of the extended storage periods. To extend the shelf life of fruits, various preservation methods have been developed, including use of ozone water, electrolyzed water, irradiation (Such as γ-ray, ultraviolet, and lighting emitting diode), physical field treatments (Such as radio frequency, magnetic fields, and electrostatic fields), modified atmosphere packaging, and coating treatments[2–5]. Although these methods offer effective preservation effects, they fail to offer real-time feedback on specific storage conditions. In comparison, the primary advantage of smart labels is their ability to provide real-time displays of freshness and offer early warnings about quality deterioration. The sensory properties of smart labels change in response to variations in environmental gas composition, temperature, and microbial load[6]. The visual changes not only assist operators at various stages of the supply chain in promptly addressing unexpected issues, but also strengthen the interaction between food and consumers[7].

Smart labels have a single or dual function of monitoring food freshness and extending shelf life when the substrate is rich in functional components, such as plant anthocyanins, antibacterial agents, and antioxidants[8]. The main method used for fabricating smart labels is casting[6]. Due to their unique structural design, digital formulation, and faster preparation, printing technology presents a viable alternative to the casting method for producing smart labels. The contact

[1]State Key Laboratory of Food Science and Resources, School of Food Science and Technology, Jiangnan University, 214122 Wuxi, Jiangsu, China. [2]China General Chamber of Commerce Key Laboratory on Fresh Food Processing & Preservation, Jiangnan University, 214122 Wuxi, Jiangsu, China. [3]Jiangsu Province International Joint Laboratory on Fresh Food Smart Processing and Quality Monitoring, Jiangnan University, 214122 Wuxi, Jiangsu, China. [4]Department of Bioresource Engineering, Macdonald Campus, McGill University, Quebec, Canada. ✉e-mail: min@jiangnan.edu.cn

area between the labels and external components can be adjusted through specialized printing structures, which better satisfies the requirements for chromogenic effects and controlled release[9]. For example, Zhou et al. [9] utilized coaxial 3D printing technology to prepare dual-functions labels containing blueberry anthocyanins and 1-methylcyclopropene, which reflected changes in freshness of litchis and prolonged their shelf life for up to 6 days. Li et al. [8] fabricated a sandwich-like film loaded with lemongrass essential oil by 3D printing technology for extending the shelf life of pork and monitoring pork freshness. However, these articles did not compare the differences in the release of functional component and preservation effects between cast and printed labels, nor did they provide quantitative analysis of the release kinetics. Additionally, the reported 3D printed labels were unable to accurately and visually monitor the critical turning points in food quality. Based on this, our team has published research exploring the differences among cast labels, dual-nozzle 3D printed labels, and coaxial 3D printed labels in monitoring the freshness and preservation effects of respiring climacteric fruits (Such as kiwi fruits, green mangoes, and persimmons)[10]. However, these labels only rely on a single-color change for discrimination, resulting in the limited monitoring effectiveness. To address this, we propose the development of 4D printed labels that, in addition to exhibiting a color response, introduces a new sensory dimension (Shape change). The dual-response mechanism of shape and color is expected to enable more accurate discrimination of fruit freshness. Furthermore, the deformation of 4D printed labels may alter its microstructure, thereby influencing the release behavior of the encapsulated essential oils and its preservation efficacy. Currently, key aspects of such pH-responsive 4D printed smart labels (Including preparation formulation, structural design, deformation mechanism, and the influence of deformation on their monitoring and preservation performance) remain unclear.

The concept of 4D printing was first introduced by Professor Tibbits at the Massachusetts Institute of Technology[11]. In the food sector, 4D printing primarily aims to enable the physicochemical property of 3D printed objects to respond to external environmental stimuli, such as pH, water, heat, and light[11,12]. For example, spraying external acidic or alkaline solutions cause discoloration in 3D printed products containing anthocyanins[13]. Dehydration (Such as hot air and microwave heating) generates a moisture gradient inside 3D printed products, leading to deformation[14]. Ultraviolet light promotes the transformation of ergosterol in 3D printed products into vitamin $D_2$[15]. Printing inks that respond to external stimuli, along with the internal design of 3D printed products, have positive implications for the implementation of 4D printing[11]. The respiratory peak of respiring climacteric fruits during storage leads to a sharp rise in relative humidity and carbon dioxide levels, thereby creating an external environment conducive to sensory changes in 4D printed labels. The functional requirements for 4D printed labels in this study are a visible colorimetric response to fruit-emitted carbon dioxide and a shape/microstructure morphing response to humidity. The structural change is specifically designed to facilitate the controlled release of essential oils based on the fruit's physiological state. Outperforming cast and 3D printed labels, the functional components in 4D printed labels are expected to exhibit the timelier color response, higher release rates, and longer release durations, while using the same printing ink.

Currently, the substrates used in smart labels are typically natural polymers (Such as starch, proteins, and celluloses) or their modified derivatives[16]. Konjac glucomannan (KGM) has been widely used in food packaging owing to its abundant reactive groups, biodegradability, high water absorption, high elasticity, and low viscosity. However, pure KGM exhibits poor printability because of its inadequate mechanical strength. In the field of food printing, external additives such as starch, carrageenan, and protein effectively improve the mechanical support ability of KGM, but this tends to reduce the sensitivity of the color developer. In our previous research, we found that KGM grafted with

methacrylic anhydride (MAKGM) met the requirements for producing printed labels[6]. According to the published reports, blueberry anthocyanins (BA) and essential oils are key components in printed labels, acting as color developers and preservatives, respectively[9]. Previously, we isolated phenolic constituents from hawthorn pomace via acid hydrolysis and integrated these compounds as supplementary colorants into BA matrix to enhance its colorimetric sensitivity and antioxidant capacity[17]. Our research also demonstrated that the microwave dry-heating method yielded a more stable essential oil emulsion in less time than the traditional method[18]. Therefore, we used the garlic essential oil (GEO) emulsion prepared by the microwave dry-heating method as the preservative. In this study, kiwi fruits, green mangoes, and persimmons were selected as typical respiring climacteric fruits for testing. Because their spoilage often occurs in hard-to-detect areas, such as inside kiwi fruits and underneath the tip of persimmons. Additionally, it is difficult for consumers to assess their freshness, which results in missing the optimal time for consumption.

There have been reports on the combination of labels and machine learning for intelligence and automation of monitoring. Although preliminary research has been conducted on this strategy for vegetables[19], fish[20] and beef[21], its application in fruits remains limited. Given the efficient performance of image processing and classification tasks on mobile devices, this study selected four lightweight deep convolutional neural network (DCNN) models: GhostNet, MobileNet, ShuffleNet, and Xception. These models were integrated with the studied labels to evaluate their monitoring accuracy. It is expected that consumers will be able to identify the freshness level of fruits by simply scanning the labels using their mobile phones.

To prepare 4D printed labels, the following work was carried out. Firstly, principal component analysis (PCA) classified the quality levels of respiring climacteric fruits during storage. The changes in relative humidity and carbon dioxide content in the fruit package were explored. Then, we investigated the effects of adding GEO emulsion and varying geometric structures (Including printing layers, filament intersection angles, and infill ratios) on the deformation of 4D printed labels. The mechanism underlying this improved performance were explained. Next, the relationship between the deformation degree and GEO release, as well as the release kinetics, was studied. Finally, the response of cast, 3D printed, and 4D printed labels in detecting the quality levels and prolonging the shelf life of respiring climacteric fruits was determined and outlined. A comparative analysis of the discriminative efficacy of four DCNN models was performed to select the best-performing label-model combination. The performance of the selected model-label pair in fruit freshness assessment was further validated. These findings demonstrate the feasibility of 4D printing technology for food preservation and offer insights for developing smart packaging that evolves from static monitoring to dynamic response.

## Result
### Analysis of MA grafted on KGM
New peaks in the 1H-nuclear magnetic resonance (1H NMR) spectra and the attenuated total reflectance-Fourier transform infrared (ATR-FTIR) spectra confirmed the success of the grafting process. As illustrated in the 1H NMR spectra (Fig. 1A), two new peaks confirmed the incorporation of methacrylate groups: a methylene signal ($=CH_2$) at 5.25-5.75 ppm and a methyl signal ($-CH_3$) at 1.50-1.76 ppm[22,23]. Furthermore, as shown in the ATR-FTIR spectra (Fig. 1B). The main characteristic peaks of these samples were observed at 3200-3400 $cm^{-1}$ (O-H stretching), 2920-2940 $cm^{-1}$ (C-H stretching), 1650-1720 $cm^{-1}$ (C = O stretching), 1490-1650 $cm^{-1}$ (C = C stretching), 1300-1460 $cm^{-1}$ (C-H in-plane bending), 1000-1300 $cm^{-1}$ (C-O stretching) and 800-960 $cm^{-1}$ (C-H out-of-plane bending of =C-H or ring structure)[6]. The appearance of new peaks (C = C stretching vibration of MA) at 1490-1650 $cm^{-1}$ in the ATR-FTIR spectra of MAKGM served as evidence for a successful

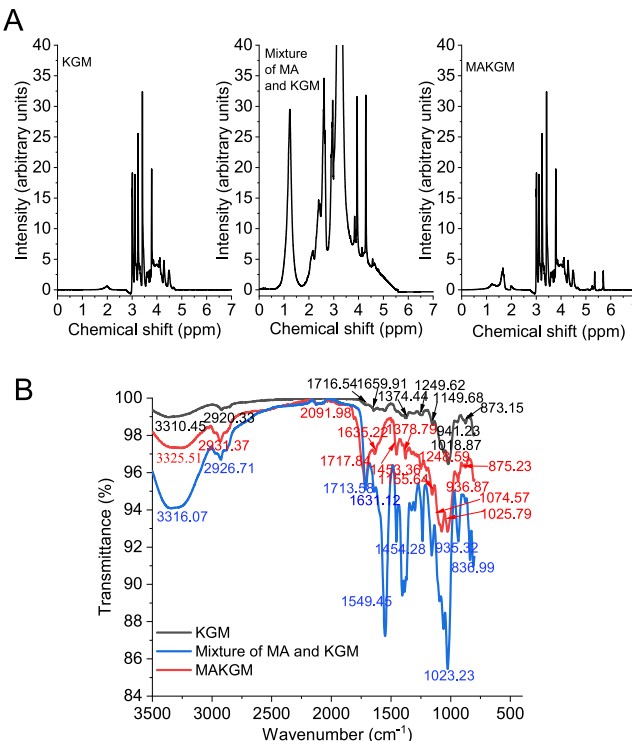

**Fig. 1 | Characterization of methacrylic anhydride (MA) grafted onto konjac glucomannan (KGM). A** [1]H NMR spectra at 600 MHz. **B** ATR-FTIR spectra collected with 32 scans per test with a resolution of 4 cm[-1] in the range of 4000 to 800 cm[-1]. $n = 3$ independent samples per group. Source data are provided as a Source Data file.

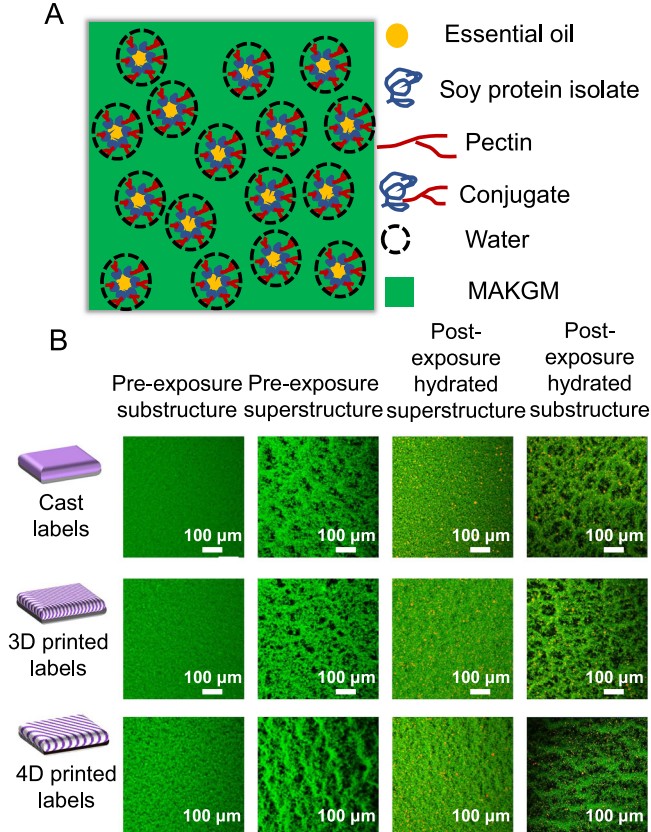

**Fig. 2 | Microstructure of different labels. A** Schematic diagram of the physical states of each component in the essential oil-loaded printing ink. **B** CLSM spectra (400 ×) of the upper and lower structures of cast, 3D printed, and 4D printed labels under different conditions. $n = 3$ independent samples per group.

interaction, as no such peaks were observed in pure KGM or the physical mixture.

## Microstructure of the upper and lower layers in the label

Based on our previous research, this experiment prepared a soy protein isolate-pectin conjugate via the Maillard reaction[18]. In the oil-water system, the soy protein isolate in the conjugate rapidly adsorbed at the oil-water interface to form an amphiphilic film around the oil droplets, while pectin readily bound with water to thicken the film because of its excellent water solubility. When MAKGM was added to the emulsion, before light exposure, MAKGM molecules absorbed water and swelled, with their molecular chains extending but exhibiting extremely weak network structure. At this stage, as shown in Fig. 2A, the water-coated emulsion droplets were entrapped within the MAKGM matrix. Confocal laser scanning microscopy (CLSM) images confirmed the differences in the network structure of the printing ink. As shown in Fig. 2B, before light exposure, the water-absorbed MAKGM molecules were densely packed without exhibiting a distinct network structure. Upon light exposure, riboflavin generated free radicals, promoting MAKGM interconnection via photosensitive groups (methacrylate groups)[24]. At this stage, MAKGM molecules began to aggregate, forming a porous network structure. Under the identical light irradiation conditions, the light-blocking effect of the label's upper layer negatively impacted the crosslinking degree of the lower structure. Among the three studied labels, owing to pore differences in the upper structure, 4D printed labels exhibited the highest crosslinking degree in its lower layer, followed by 3D printed labels and cast labels. This trend was consistently reflected in the aggregation state of MAKGM (Fig. 2B). Due to the weak intermolecular interactions between the emulsion droplets and MAKGM, the droplets failed to migrate with MAKGM during its aggregation, resulting in their presence within the pores of the

network structure. This phenomenon was more pronounced in the CLSM images of the lower-layer structure of 4D printed labels.

## Effect of GEO contents on deformation degree

For 4D printed products, the degree and direction of deformation generated by water stimulation depend on hydrophilic differences and spatial arrangement of two or more printing inks[12]. Various amounts of GEO emulsion were introduced into the lower structure to create a hydrophilicity difference between the two layers of the label. As shown in Figs. 3, 4D printed labels gradually reached their maximum curl on 4th day at 93% relative humidity and 25 °C. After 4th day, the curly shape gradually stretched out over time. In contrast, the label without GEO emulsion exhibited no obvious curl. When the addition of GEO emulsion increased from 0% to 10% (w/w), the average curvature increased by 0.74 cm[-1] (Fig. 4A). However, compared with the label containing 10% (w/w) emulsion, the further increase in emulsion addition led to a decrease in average curvature of 0.42 cm[-1] (Fig. 4A). Generally, addition of more hydrophobic materials (i.e., GEO emulsion) reduces the substrate's hydrophilicity, which is favorable for the deformation due to hydrophilic differences[12]. However, printed labels containing more than 10% (w/w) GEO emulsion showed the reduced curl. To explain this contradictory phenomenon, we subsequently examined the physicochemical differences between the upper and lower structures of printed labels.

## Reasons for deformation diversity of labels with different GEO contents

High printing fidelity is essential to prevent the mixing of inks between the upper and lower structures while meeting the structural design

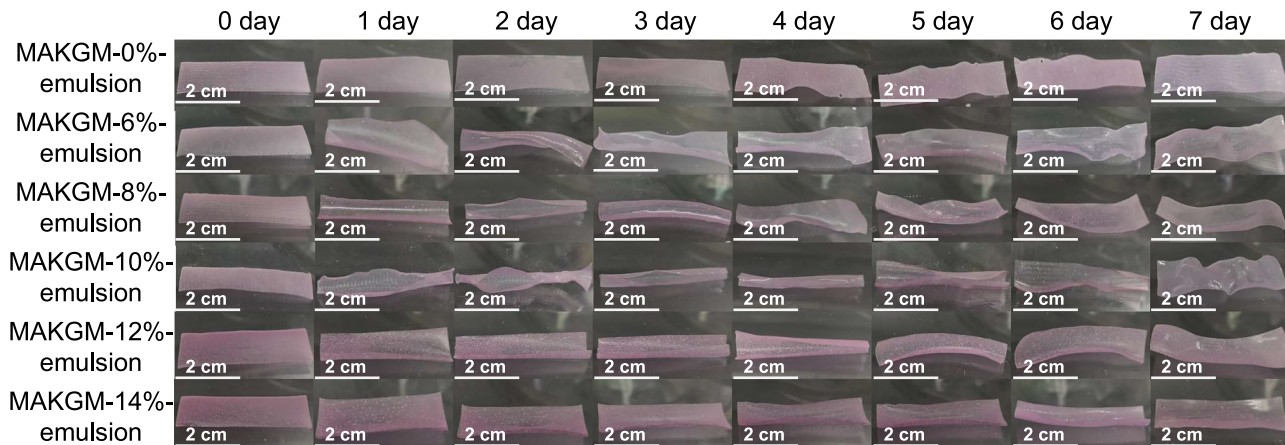

**Fig. 3 | Effect of the essential oil content on deformation degree.** All tests were conducted under controlled conditions of 93% relative humidity and 25 °C. The lower structure of all labels contained 10% (w/w) emulsion and 0.08% (w/v) riboflavin, while the upper structure consisted of 0.2% (w/v) color developer and 0.08% (w/v) riboflavin. The infill ratio was set to 100% for the lower structure and 30% for the upper structure. The filament intersection angle was fixed at 60°. MAKGM is konjac glucomannan grafted with methacrylic anhydride. $n = 3$ independent samples per group.

requirements. The degree of fidelity was assessed by comparing the similarity between the printed labels and the designed hollow cylinder. As shown in Fig. 4B, the printing fidelity of the ink without emulsion was 88.02%, and the addition of anthocyanins had no effect on the printing results. In comparison, the printing fidelity decreased by 8.82–40.21% as the emulsion concentration increased from 6% to 14% (w/w). The difference in fidelity was related to the mechanical strength of printing inks. The strong mechanical strength alleviated the line expansion and collapse caused by gravitational loads[25]. Supplementary Fig. 1 presents the results of frequency scanning of the printing inks containing different GEO emulsion before and after light exposure. The results indicated that as GEO emulsion increased, the storage modulus of the printing inks gradually decreased (Supplementary Fig. 1A, C), while the loss modulus progressively increased (Supplementary Fig. 1B, D). The Tanδ value fell within the range of 0 to 1. A value closer to 0 indicated that the printing ink exhibited higher elasticity, while a value closer to 1 signified higher viscosity. As shown in Supplementary Fig. 1E, F, with the increase in the content of GEO emulsion, the Tanδ value of the samples gradually increased, indicating a reduction in the elasticity and an enhancement in the viscosity of the samples, which was unfavorable for improving their mechanical properties. Additionally, compared with the samples before light exposure, the Tanδ value of the same samples decreased after light exposure, suggesting that the light-treated samples exhibited more solid-like behavior, which macroscopically manifested as an improvement in the mechanical properties of the samples. The addition of emulsion had an inverse effect on the mechanical strength, suggesting a weakened gel network structure.

The water content of the single-layer film in a high-humidity environment was measured to evaluate its water absorption capacity. As shown in Fig. 4C, the emulsion addition significantly affected the water content of the single-layer film. During the testing period, the water content of all films ranged from 12.71% to 103.89% (w/w). Based on the results in Fig. 3, greater differences in moisture content between samples with and without emulsion were associated with increased deformation. For example, 4D printed labels containing 10% (w/w) emulsion in Fig. 3 had the maximum degree of deformation on 4th day. At this point, the difference in moisture content between the upper and lower structures of this label was about 34% (w/w). All 4D printed labels displayed in Fig. 3 demonstrated a smaller degree of deformation on 7th day, while the difference in moisture content between the upper and lower structures of all labels was less than 8% (w/w).

The contact angle was measured to evaluate the surface hydrophobicity of the single-layer film printed using different formulations. As shown in Fig. 4D, E, the addition of anthocyanins did not affect the change of contact angle because of the small quantity. When the emulsion addition was less than 10% (w/w), the surface hydrophobicity was proportional to the emulsion content within 120 s. However, when it exceeded 10% (w/w), an inverse relationship between the surface hydrophobicity and the emulsion content was observed after about 30 s. Among all the samples, the sample containing 10% (w/w) emulsion demonstrated the best surface hydrophobicity. The hydrophobicity of the printing ink itself and the density of the formed network structure affect the water absorption capacity of printed labels[26]. Based on the fact that adding too much hydrophobic emulsion instead increased the water absorption capacity (For example, samples with more than 10% (w/w) emulsion), it was speculated that too much emulsion had a negative impact on the density of the network structure. For the printed double-layer label, the microstructure of its lower structure before deformation and at maximum deformation was determined using scanning electron microscopy (SEM). As shown in Fig. 5A, for the lower structure before deformation, its pore size increased along with the emulsion addition, resulting in a decrease in the density of the network structure. The loose network structure reduced the obstruction of water migration, making the labels more susceptible to water infiltration. This confirmed the hypothesis that excess emulsion (More than 10%, w/w) had a negative effect on deformation. Additionally, compared with the lower structure before deformation, the pore size increased after deformation, and the pore shape transitioned from round to oval. This implied that the deformation effect altered the network structure. It is worth noting that the lower structure containing more than 10% (w/w) emulsion exhibited roughness and fracture in the network structure after reaching the maximum deformation. This is ascribed to the fact that the weak intermolecular forces within the network structure are insufficient to withstand the pulling forces generated by deformation, leading to the structural damage[27]. Figure 5B demonstrates the variation in total porosity of printing inks containing different emulsion contents before and after deformation. The results indicated that the total porosity increased with the increasing emulsion content (Ranging from 19% to 64%). Compared with the pre-deformation state, the total porosity after deformation exhibited a 20–46% increment.

The lower structure of 4D printed labels not only required strong hydrophobicity but also needed a robust network structure to resist pulling forces, thereby preventing the rapid water infiltration caused

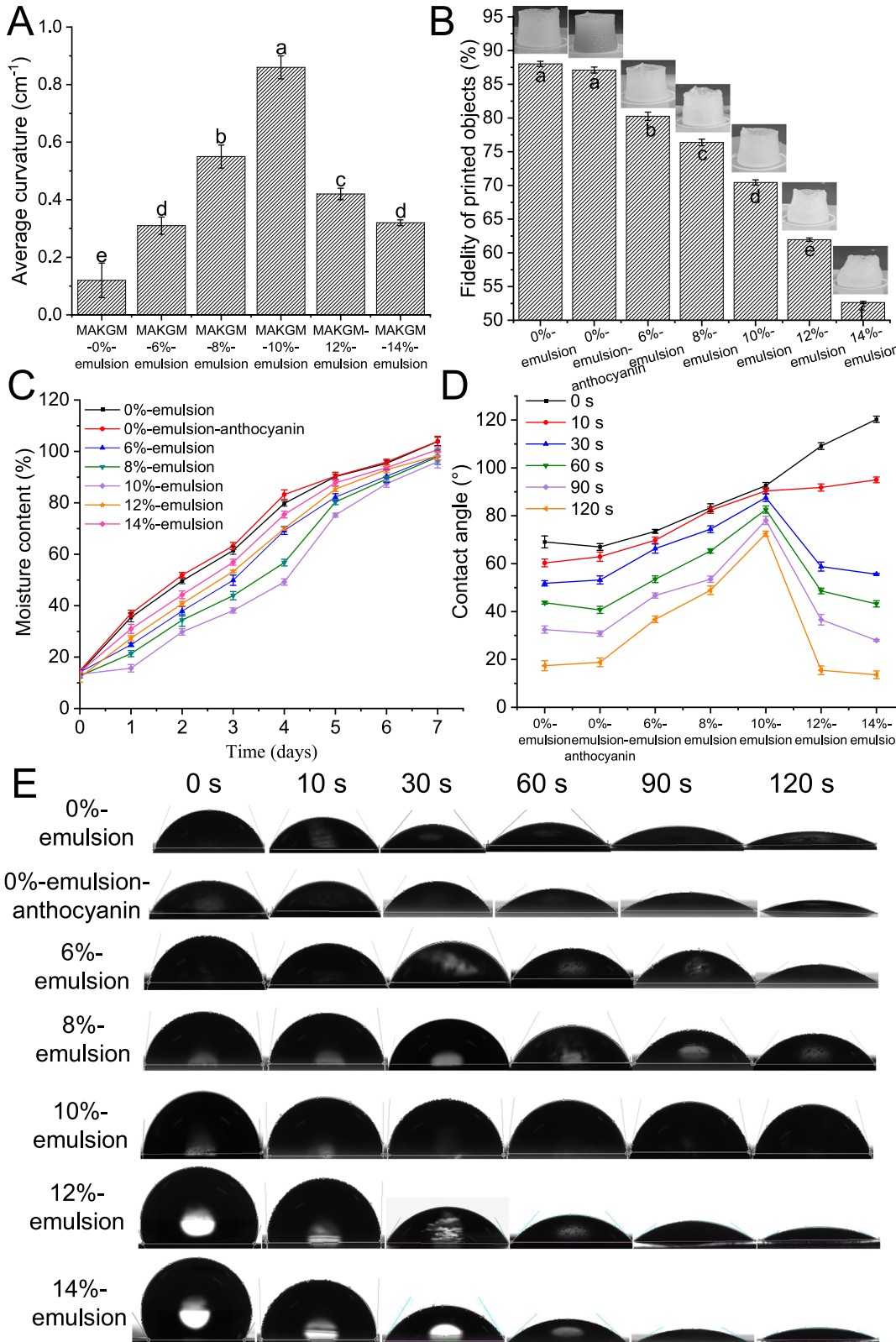

**Fig. 4 | Comparison of macroscopic physicochemical properties. A** Average curvature of 4D printed labels with the maximum deformation as shown in Fig. 3. **B** Fidelity of printing inks with different formulations. **C** Changes in moisture content of the single-layer structure with different formulations. **D, E** Contact angle of the single-layer structure with different formulations. MAKGM is konjac glucomannan grafted with methacrylic anhydride. All data in Fig. 4A–D are mean ± S.D. $n = 5$ independent samples per group for contact angle measurements, and $n = 3$ independent samples per group for all other tests. Different letters indicate statistically significant differences between samples ($p < 0.05$). Source data are provided as a Source Data file.

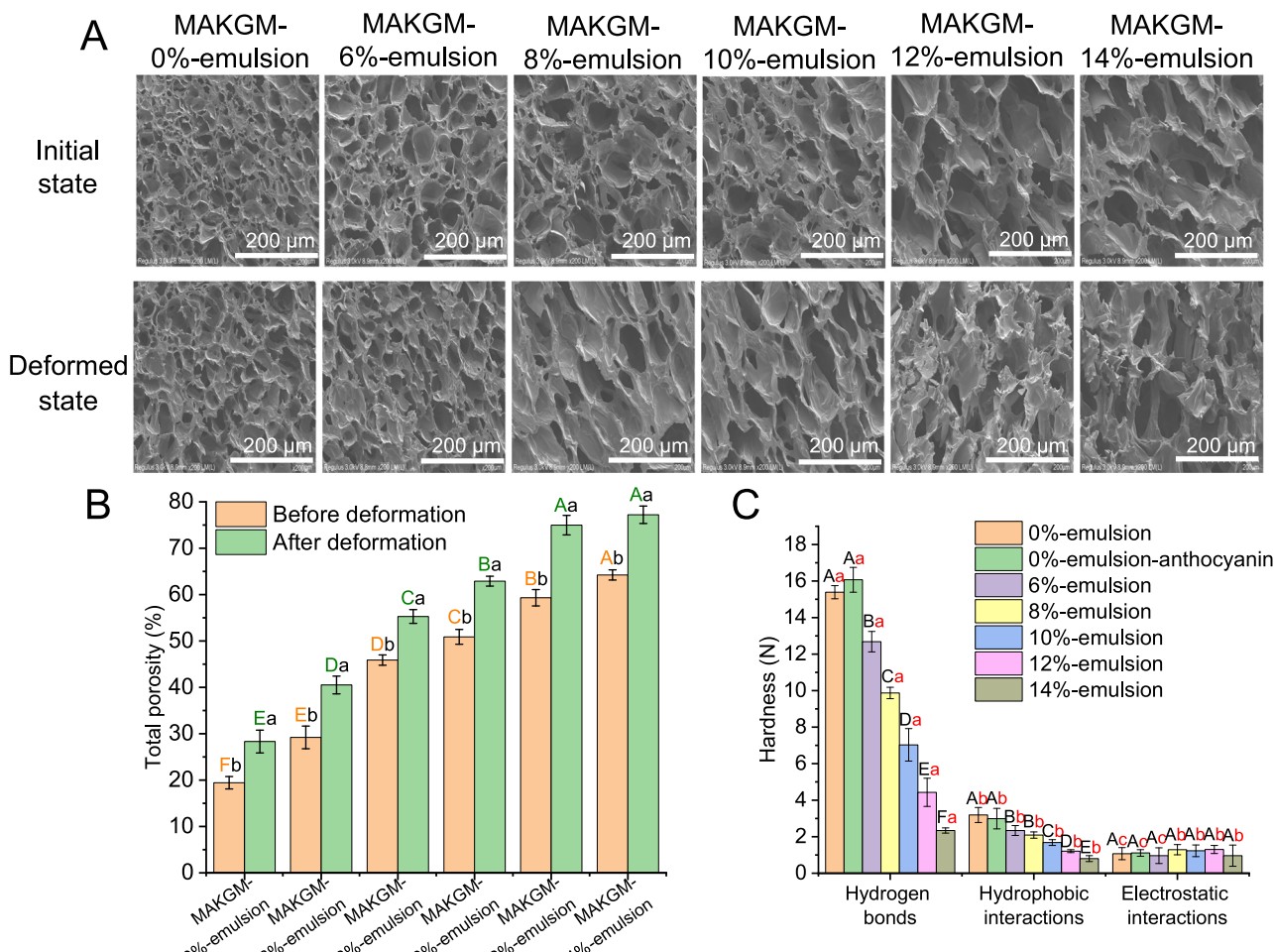

**Fig. 5 | Comparison of microscopic physicochemical properties. A** SEM images (200 ×) of 4D printed labels with the maximum deformation as shown in Fig. 3. **B** Total porosity of the lower structure of printed labels without deformation and with the maximum deformation. **C** Intermolecular force of printing inks with different formulations after light treatment. MAKGM is konjac glucomannan grafted with methacrylic anhydride. All data in Fig. 5B, C are mean ± S.D. $n = 3$ independent samples per group. Different capital letters indicate statistically significant differences ($p < 0.05$) between samples for a given measured index, while different lowercase letters denote significant differences ($p < 0.05$) within the same sample across different measured indices. Source data are provided as a Source Data file.

by structural damage. As shown in Fig. 5C, hydrogen bonds and hydrophobic interactions were the dominant molecular forces, and they gradually weakened with the increased emulsion content. For example, compared with the sample containing 10% (w/w) emulsion, the sample containing 14% (w/w) emulsion showed a 61.19% reduction in hydrogen bonds and a 47.02% reduction in hydrophobic interactions. This indicated that the addition of emulsion had a negative impact on molecular forces, which was consistent with the results observed by SEM (Fig. 5A).

In summary, the successful deformation of 4D printed label developed in this study required high printing fidelity, variations in hydrophilicity, and a network structure capable of resisting the pulling forces caused by deformation. A low emulsion concentration was insufficient to create significant hydrophilic differences, while a high emulsion concentration compromised the integrity of the network structure. For subsequent studies, a 10% (w/w) emulsion was employed to fabricate cast, 3D printed, and 4D printed labels.

### Effect of geometric designs on deformation degree

As shown in Fig. 6, this study investigated the influence of three key geometric designs on deformation. Various geometric design led to the macroscopic differences in the degree of deformation. As shown in Fig. 6A, the labels reached their maximum curvature on 4[th] day and

subsequently began to unfold. To intuitively demonstrate the shape change, the supplementary video captured the complete shape morphing process of 4D printed labels in an acidic phosphate buffer solution (pH = 4), presenting the dynamic process from curling, through reaching maximum curvature, to unfold. This phenomenon arose from the water migration rates in the upper and lower structures[28]. At the initial stage (On days 0 ≤ t < 1), the upper structure of the label was only partially inflated by water absorption, and the resulting pulling force was difficult to drive the lower structure. As water continued to migrate into the interior of the upper structure, the label began to curl (On days 1 ≤ t ≤ 4). When the lower structure was gradually saturated with water (On days 4 < t < 7), the curly labels began to unfold because of the decreased water gradient between the upper and lower structures. The maximum curvature remained unchanged at 0.86 cm⁻¹ when the number of upper layers was ≤2. Exceeding the critical value led to a 25.53%-38.82% decrease in the maximum curvature (Fig. 6B). As shown in Fig. 6A, changing the filament intersection angles caused different shapes of deformation. In previously reported studies on 4D food deformation caused by dehydration, the shape change was perpendicular to the direction of filaments of the driving layer[29,30]. This study found the shape change stimulated by water also followed this rule. Additionally, the shape of deformation affected the value of curvature. Compared with the

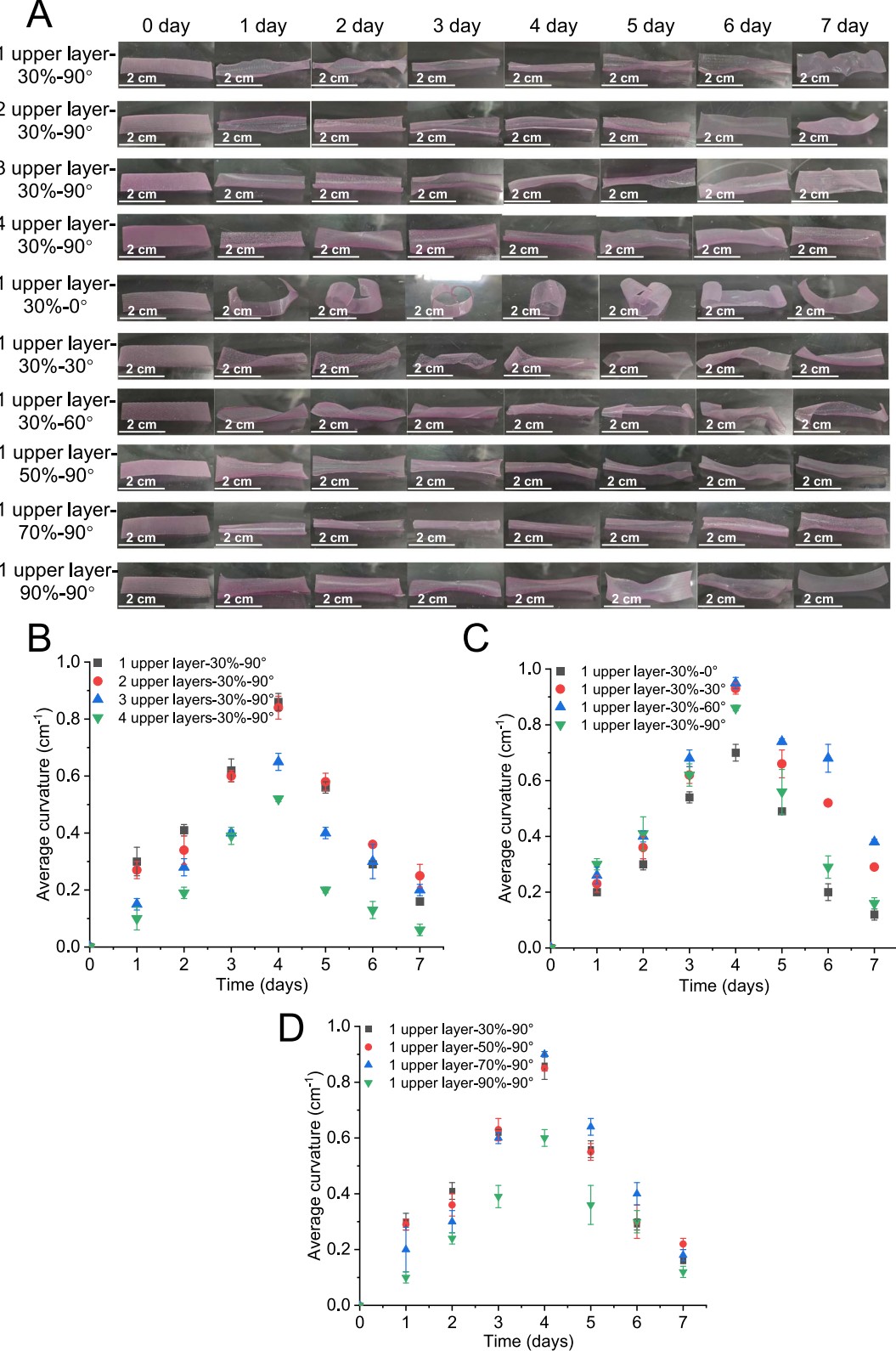

**Fig. 6 | Effect of structural designs on deformation degree. A** Shape changes of 4D printed labels at 25 °C and 93% relative humidity. **B** Average curvature of 4D printed labels with printing layer number of the upper structure from 1 to 4. **C** Average curvature of 4D printed labels with filament intersection angles from 0° to 90°. **D** Average curvature of 4D printed labels with infill ratios from 30% to 90%. All data in Fig. 6B–D are mean ± S.D. $n = 3$ independent samples per group. Source data are provided as a Source Data file.

maximum curvature (0.86 cm⁻¹) of the 1 upper layer-30%-90° label, the curvature increased by 8.14%-10.47% as the filament intersection angles varied between 30° and 60° (Fig. 6C). While it approached to 0°, the curvature decreased by 18.60% (Fig. 6C). As shown in Fig. 6D, when the number of the upper structure was one layer, the filament intersection angle was 90° and the infill ratio was within 30%-50%, the maximum curvature had no significant change (0.86 cm⁻¹). When the infill ratio increased to 70%, the maximum curvature increased by 4.7% because the increased infill ratio was beneficial for generating greater swelling force after water absorption. However, when it reached 90%, the maximum curvature decreased by 30.23%. A further increase in the infill ratio reduced the porosity of the upper structure, which in turn slowed the water migration rate. This resulted in a prolonged swelling time for the driving layer and an extended water permeability period for the constraining layer.

The degree of deformation resulted in varying levels of structural damage to the restraint structures of the printed labels. As shown in Fig. 6, all printed labels maintained a higher degree of deformation on days 3-5. During this period, the average curvature was ranked in descending order as follows: 1 upper layer-30%-60° (0.79 cm⁻¹) > 1 upper layer-30%-30° (0.74 cm⁻¹) > 1 upper layer-70%-90° (0.70 cm⁻¹) > 1 upper layer-30%-90° (0.68 cm⁻¹) = 1 upper layer-50%-90° (0.68 cm⁻¹) = 2 upper layers-30%-90° (0.68 cm⁻¹) > 1 upper layer-30%-0° (0.58 cm⁻¹) > 3 upper layers-30%-90° (0.48 cm⁻¹) > 1 upper layer-90%-90° (0.41 cm⁻¹) > 4 upper layers-30%-90° (0.34 cm⁻¹). To verify the effect of deformation on structural damages, Fig. 7A shows the SEM images of the lower structure of all printed labels on 5th day. SEM images indicated that compared with the undeformed lower structure, the pore size of the deformed lower structure increased, and their shape changed from round to oval or even collapse because of the swelling force. Furthermore, the ranking of pore size and degree of structural damage, from high to low, aligned with the previously mentioned order.

In summary, geometric design changed the direction and degree of deformation, resulting in various degrees of damage to microstructures. This suggested a positive effect on the release of GEO in the lower structure.

## Effect of geometric designs on GEO release

The effect of three key geometric designs on the release profiles of GEO was investigated. As illustrated in Fig. 7B, three release phases (i.e., on days 0-1, 1-8, and 8-10) were observed. The first stage matched the rapid release phase because of the shortest diffusion distance and largest contact surface, followed by a slowly release due to the hindrance of barrier layers, until a constant value was reached[27]. Printed labels with larger deformation, such as both 1 upper layer-30%-30° and 1 upper layer-30%-60° labels, had higher cumulative release levels. This was due to the altered microstructure during deformation which significantly reduced the diffusion pathway and binding force of the network structure[11]. For example, the cumulative release of 1 upper layer-30%-60° labels was 42.32% higher than that of 4-upper layer-30%-90° labels on 10th day. Therefore, employing deformation to control GEO release was considered feasible.

The Ritger-Peppas model was applied to the release data in order to investigate the GEO release behavior. $R^2$ value (0.9843-0.9960) suggested that this model was satisfactory to describe the release characteristics of GEO. As illustrated in Fig. 7C, the values of the diffusional exponent (n) were $0.43 \leq n < 0.85$. In other word, GEO release followed the mechanism of non-Fickian or anomalous transport, which indicated that three-dimensional morphology of the carrier influenced the GEO release[26]. Furthermore, a higher kinetic constant (k) represented a higher free volume, reflecting that GEO was subjected to smaller binding forces during the release process, which was beneficial for a higher amount of GEO release[22]. In view of the cumulative GEO release content, 1 upper layer-30−60° was selected for further study.

## Analysis of graded quality of respiring climacteric fruits

Changes in some key quality indicators of kiwi fruits, green mangoes, and persimmons during storage are shown in Supplementary Fig. 2. Hardness, titratable acidity, total soluble solids, and weight loss of the three fruits showed an initial slow change, followed by a rapid increase or decrease phase. Because respiring climacteric fruits need to be harvested before maturity, as their enzyme activity (Such as pectinases, cellulases, amylases, proteases, and oxidases) is low at this stage, resulting in a slow change in quality[27]. The rapid decrease/increase phase is related to the respiration peak of fruits (Supplementary Fig. 3A–C). During this time, increased enzyme activity facilitates the separation of pectin and cellulose, leading to an increase in soluble pectin and a decrease in hardness (Supplementary Fig. 2A, D, G); metabolic breakdown of starch and organic acids results in the accumulation of total soluble solids (Supplementary Fig. 2B, E, H) and sweetness (Supplementary Fig. 2A, D, G); and strong transpiration along with respiration increases weight loss (Supplementary Fig. 2B, E, H)[23]. As shown in Supplementary Fig. 2A, D, G, the overall acceptance and taste of fruits were accepted by consumers near their respiratory peak. However, due to aging of fruits and proliferation of microorganisms in the later stage of storage, fruits gradually decayed, resulting in a decrease in sensory scores.

The principal component analysis (PCA) results presented in Supplementary Fig. 2C indicated that kiwi fruits were categorized into four quality stages based on harvest time: days 0-–5 (immature), days 6–8 (edible), days 9–11 (sub-edible), and days ≥ 12 (spoiled). Green mangoes followed a similar classification into four stages: days 0–7 (immature), days 8–11 (edible), days 12–14 (sub-edible), and days ≥ 15 (spoiled) (Supplementary Fig. 2F). Persimmons were grouped into three stages: days 0–23 (immature), days 24–28 (edible), and days ≥ 29 (spoiled) (Supplementary Fig. 2I).

## Analysis of changes in environmental conditions caused by fruits

From Supplementary Fig. 3A, B, C, it can be observed that all three selected fruits exhibited a significant respiration peak during storage, with a maximum respiratory rate between 18 and 21 mg/kg*h. As the respiratory rate changed, the volume fraction of carbon dioxide and relative humidity in the plastic box were also affected. Compared with the initial stage, the carbon dioxide content increased by 50%-60%, while the relative humidity level rose by 15%-19% in the later stage of storage. These changes created favorable environmental conditions for discoloration and deformation of 4D printed labels.

Carbon dioxide in the environment dissolves in water to form carbonic acid, which plays a crucial role in discoloration of the color developer. Supplementary Fig. 3D shows that the pH value of the solution decreased from 7.0 to 4.0 as the volume of the injected carbon dioxide increased from 0% to 100% (v/v). Accordingly, carbon dioxide generated by the fruits reduced the solution's pH value to around 5.0 (Supplementary Fig. 3E). As can be seen from Supplementary Fig. 3E, F, the color developer exhibited different color reactions within the pH range from 7.0 to 5.0, transitioning gradually from purple to red. This established a basis for the label to respond to fluctuations in external carbon dioxide levels.

## Implementation of monitoring freshness using cast and printed labels

Considering that the substrate used in the labels might affect discoloration, the label was tested on the real respiring climacteric fruits to observe any changes. The effects of cast, 3D printed, and 4D printed labels on monitoring and controlling the quality changes of respiratory climacteric fruits were investigated. Supplementary Fig. 4 indicated that there was no significant macroscopic difference between the samples containing cast and 3D printed labels as well as the control group. In other word, these two types of labels did not significantly

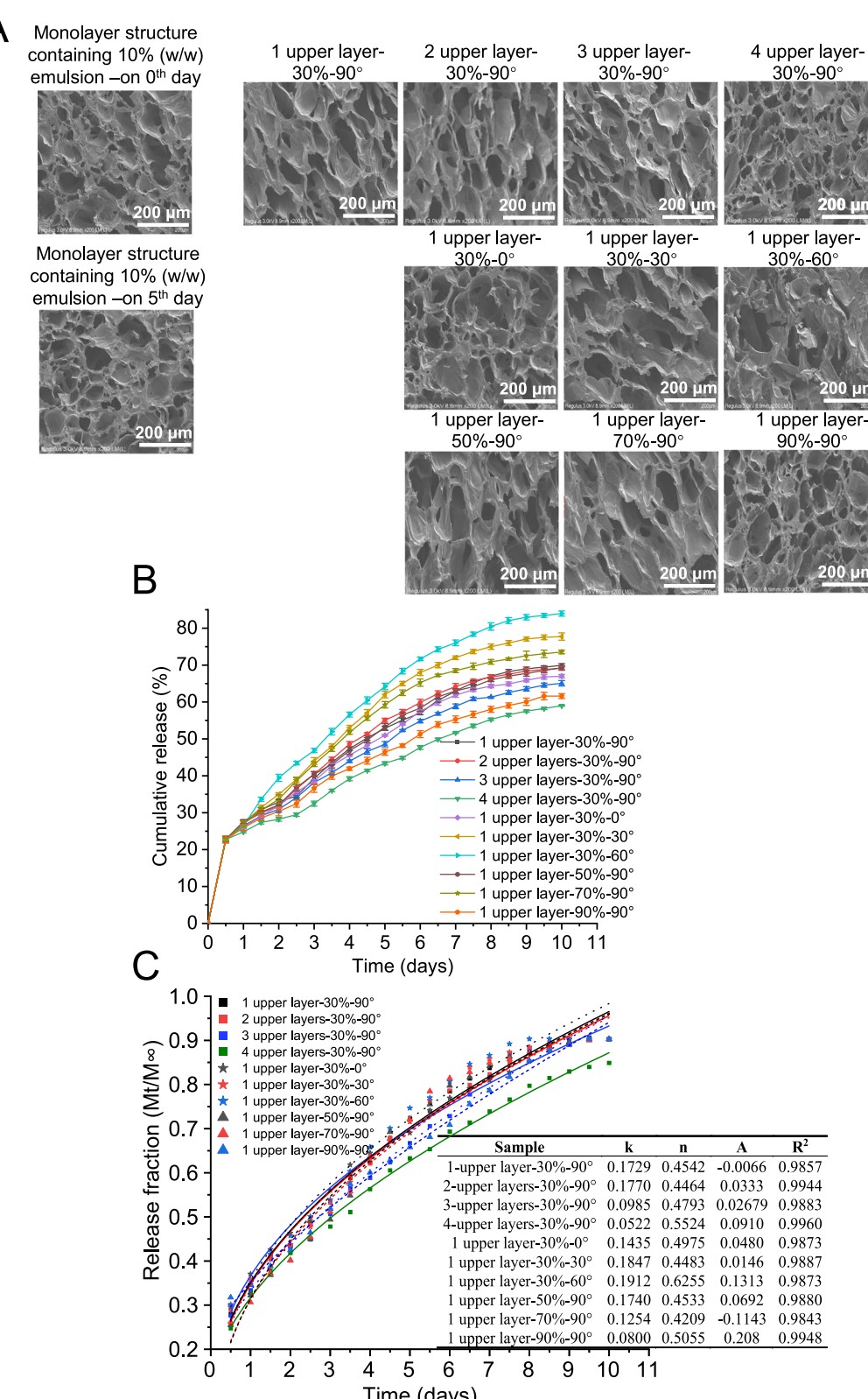

**Fig. 7 | Microarchitecture of 4D printed labels and controlled release of essential oil. A** SEM images (200 ×) of 4D printed labels after 5 days of treatment at 25 °C and 93% relative humidity. **B** Cumulative release of essential oil included in 4D printed labels at 25 °C and 93% relative humidity. **C** The result of Ritger-Peppas model fitting. All data in Fig. 7B and C are mean ± S.D. n = 3 independent samples per group. Source data are provided as a Source Data file.

extend the shelf life of fruits. In contrast, samples containing 4D printed labels consistently maintained good quality throughout the observation period.

One of the main reasons for fruit spoilage was the increase in load of microorganisms. Figure 8A–C show the trend in the total bacterial count during the storage of fruits. At a fixed time, the total bacterial count of the three selected fruits was ranked from low to high as 4D printed labels, 3D printed labels, cast labels, and the control group. This order was consistent with the cumulative amount of GEO release (Fig. 8A–C). This suggested that, compared with the casting method, printing technology increased porosity through geometric design to achieve the goal of releasing more GEO. Similar findings have been validated across food science, medicine, and related disciplines[22]. However, in the actual preservation process, 3D printed labels failed to show any advantages over cast labels. 4D printed labels are an extension of 3D printed labels. The cumulative content of GEO release from 4D printed labels was 15.84%-35.02% higher than that from 3D printed labels, and the total bacterial count was less than 13.33%-17.39% (Fig. 8A–C). As mentioned above, 4D printed deformation labels caused damage to the internal structure, and triggered the release of more GEO, which effectively inhibited microbial growth.

Compared with 3D printed labels, 4D printed labels not only had better controlled GEO release, but also provided superior visual monitoring effects. As shown in Fig. 8a–c, the color of all labels gradually lightened from purple and then changed to red. The drawback was that cast and 3D printed labels could not clearly visualize the critical point of fruit quality conversion, for example, the quality difference of kiwi fruits between days 8 and 9, green mangoes between days 11 and 12, and persimmons between days 28 and 29. This limitation was attributed to the following points: (1) The pH difference at the turning point of the fruit quality level was not significant. (2) Compared with the aqueous solution, the printing ink weakened the color intensity. (3) The printed labels required time to absorb external moisture and carbon dioxide. In contrast, the deformation degree of 4D printed labels gradually increased (or then unfolded in the plastic box containing mango or persimmon), demonstrating noticeable differences at the turning points of quality levels. This indicated a better visual monitoring effect for the quality assessment of fruits. Additionally, 4D printed labels exhibited a darker color because of the curling effect.

In summary, compared with cast and 3D printed labels, 4D printed labels presented a unique ability to respond to changes in external moisture and carbon dioxide content. 4D printed labels underwent color and shape changes, which allowed them to more accurately visualize the different quality levels of respiring climacteric fruits. Furthermore, 4D printed labels demonstrated better preservation effects.

### Implementation of monitoring fruit freshness using labels integrated with machine learning

This study evaluated four lightweight DCNNs (GhostNet, MobileNet, ShuffleNet, and Xception) with the objective of developing an accurate, efficient, and automated system for fruit freshness prediction on mobile platforms. Through extensive iterative training, these DCNN models learned to predict freshness by effectively extracting features from label images. As shown in Fig. 9A, the training loss exhibited a decreasing trend with an increasing number of iterations. The accuracies of the training results of cast, 3D printed and 4D printed labels were in the range of 83.90–87.90%, 85.65–89.40%, and 90.05–97.90%, respectively. Therefore, among the three tested labels, 4D printed labels achieved the highest training accuracy. This was attributed to the multisensory changes in both color and shape exhibited by 4D printed labels, which provided superior cues for model recognition and classification, compared with cast and 3D printed labels.

Furthermore, the four DCNN models exhibited descending compatibility with 4D printed labels: MobileNet > GhostNet > ShuffleNet > Xception. As shown in Fig. 9B, the confusion matrix diagonal shows correctly classified instances, while off-diagonal elements indicate misclassifications. In the evaluation of 720 test images of 4D printed labels (Including the three fruits) using the four DCNN models, the most misclassifications occurred between the "edible" and "sub-edible" categories. The prediction accuracy of GhostNet, MobileNet, ShuffleNet, and Xception models was 92.65%, 97.90%, 90.15%, and 90.05%, respectively. In summary, the integration of the MobileNet model with 4D printed label imagery provided an optimal approach for tracking and categorizing the fruit freshness.

## Discussion

The color change of the labels is attributed to the sensitivity of the anthocyanin structures to pH changes. The carbon dioxide produced by the fruits changed the neutral environment inside the plastic box to an acidic one. The structure of anthocyanins shifted from quinoid alkali (purple) to methanol off base (colorless), chalcone (colorless), and flavylium cations (light pink)[31].

The deformation of 4D printed labels mainly rely on the swelling mismatch between the upper and lower layers. This swelling mismatch requires the coordinated regulation of the printed formulation and the printed structure. In terms of the printed formulation, KGM grafted with MA was utilized as the printed substrate. GEO emulsion was distributed in the network formed by MAKGM, and caused a difference in hydrophilicity between the upper and lower structures of 4D printed labels. Consequently, compared with the lower structure, the upper structure exhibited a higher water absorption rate and swelling force, which generated a driving force for deformation. Additionally, printed layers and infill ratios also affected the swelling difference between the upper and lower structures by altering the contact area and the migration rate of water molecules. Infill angles and infill ratios of the upper structure influenced the direction of the driving force. Additionally, compared with cast and 3D printed labels, the network structure of 4D printed labels exhibited a more distinct longitudinal or transverse trend. This is because the printed lines of the upper structure of 4D printed labels have less adhesion, which contributes to reducing the interactions among the printed lines. The upper structure was constrained by the lower structure during the swelling process. Under the combined effect of swelling force and constraining force, the direction of deformation was perpendicular to the printing path of the upper structure. What's more, the network structure of the lower layer of 4D printed labels also changed under the action of the swelling force of the upper structure, which facilitated the release of essential oils.

Compared with cast and 3D printed labels, 4D printed labels offer some key advantages in monitoring fruit freshness and preserving fruit quality. First, the deformation causes an overlap of 4D printed labels, which helps visually deepen the color. Secondly, the structural damage caused by deformation is more conducive to the release of preservatives, thereby extending the shelf life of fruits. Thirdly, the deformation of 4D printed labels enhances the ability of machine learning models to more accurately and easily assess the freshness levels of fruits, particularly at critical quality transition points. The reduction in economic losses far outweighs the cost of label production (Approximately $0.04 per kilogram of fruits).

A fundamental limitation of our study is that, as shown in Supplementary Fig. 5, a dedicated space needs to be designed in the food packaging in order to protect the label from damage by fruits and allow consumers to easily observe the label. Another limitation is that the application of 4D printed labels in this study is restricted to respiring climacteric fruits, and the potential for their use in other fresh produce sectors has not been explored. Meat and seafood tend to accumulate volatile amines, hydrogen sulfide, and water vapor

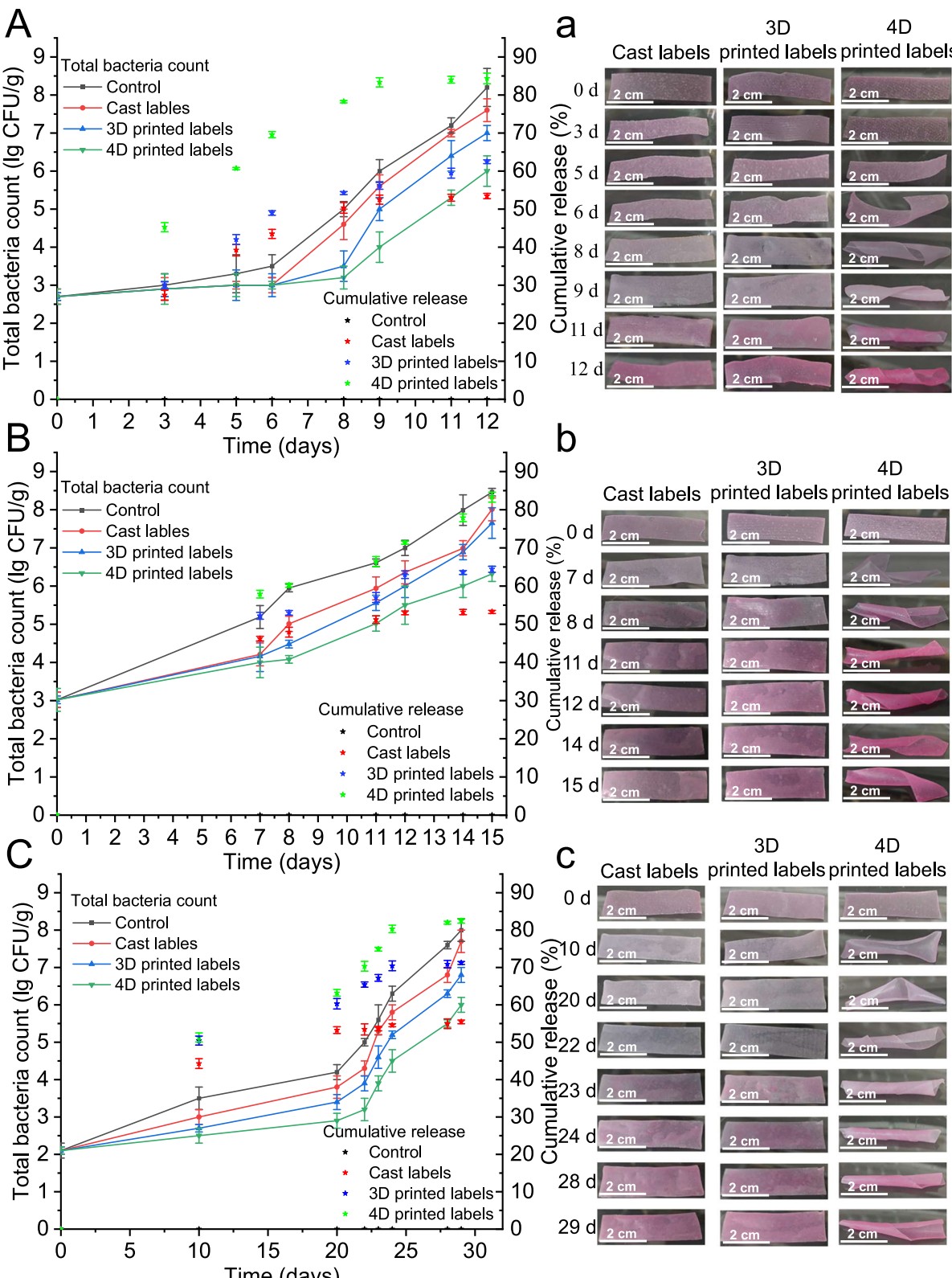

**Fig. 8 | Antimicrobial performance and sensory changes of different labels in practical applications.** Microbial changes in fruits and essential oil release from labels during storage at 25 °C: **A** kiwifruits, **B** green mangoes, and **C** persimmons. The color and shape changes of **a** kiwi fruits, **b** green mangoes, and **c** persimmons at the turning point of fruit quality changes. All data in Fig. 8A–C are mean ± S.D. n = 3 independent samples per group. Source data are provided as a Source Data file.

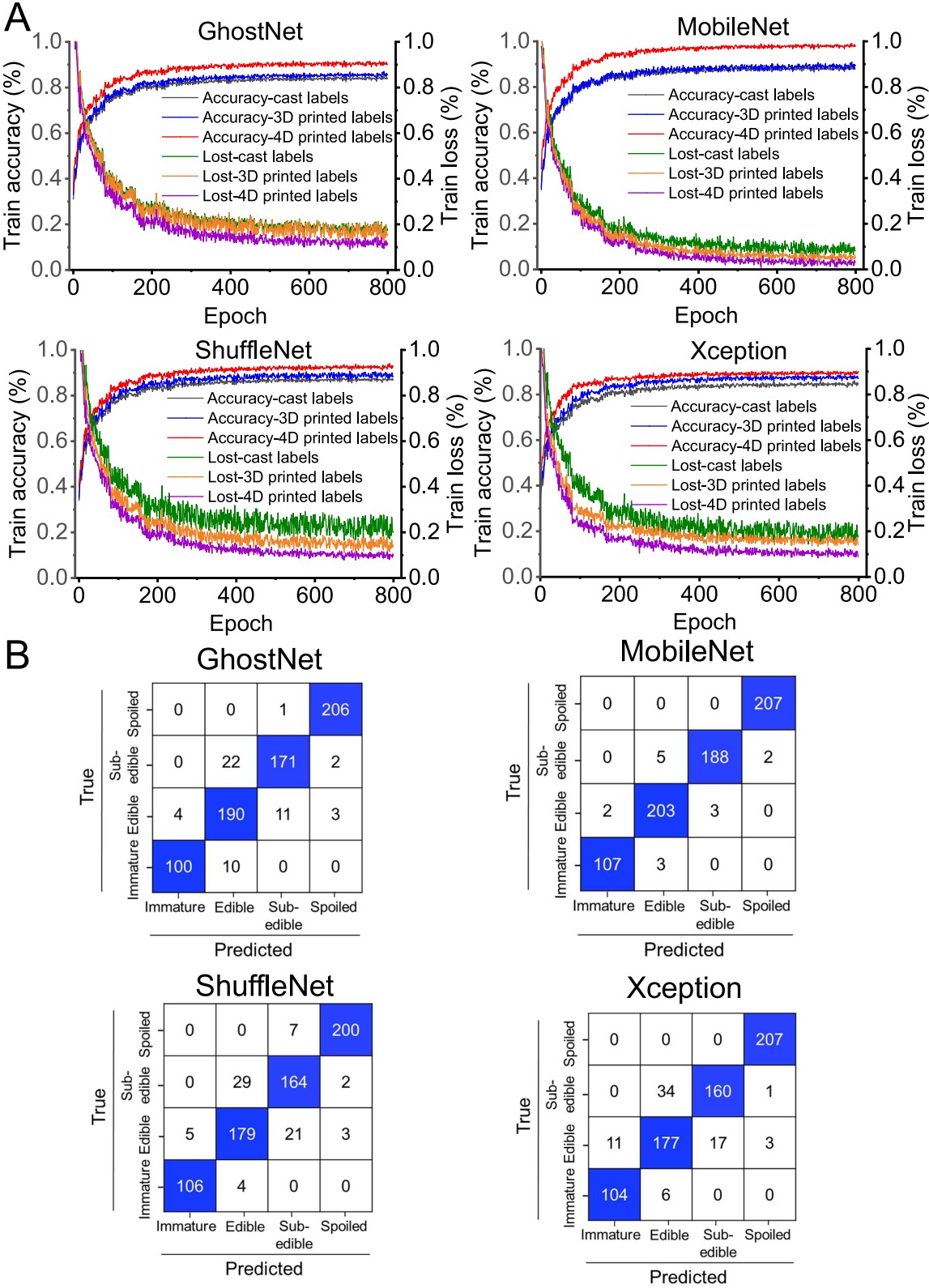

**Fig. 9 | Monitoring the freshness of respiring climacteric fruits using via smart labels integrated with deep convolutional neural network (DCNN). A** The training results of four lightweight DCNN models combined with cast, 3D printed, and 4D printed labels. **B** The prediction performance of four lightweight DCNN models combined with 4D printed labels. Source data are provided as a Source Data file. Code is available in the Supplementary Code package.

during storage, which changes the pH value and relative humidity inside the food package. Based on this, 4D printed labels developed in this study may be suitable for monitoring their freshness and preservation. In addition, 4D printing technology is also used to develop

other types of labels with multi-source information detection, such as pH-temperature responsive labels. As shown in Supplementary Fig. 6, the flower-shaped printed labels not only change color in response to pH variations, but also bloom to different degrees in response to

temperature stimuli. These pH-temperature responsive labels might monitor food freshness while reducing the occurrence of temperature breaks in the supply chain of fresh food.

The next phase of our research will explore the key sensing signals related to the freshness transition of fresh foods, develop new sensitive sensing materials, or utilize machine learning to combine various sensing materials to form an effective perception array. By leveraging the advantages of 4D printed labels in discoloration and deformation, we aim to expand the application of 4D printed labels in monitoring the freshness and preservation of fresh foods.

The industrial application of 4D printed labels requires identifying the specific characterization indicators for quality deterioration of fresh foods, and clarifying the intrinsic relationships between these indicators and key signal perception, as well as quality degradation. It is essential to establish standards for categorizing the freshness of fresh foods. While 4D printing technology has developed automated production pathways for smart labels, there is currently no integrated device capable of producing printed labels. This limitation poses challenges for achieving large-scale production. Therefore, it is essential to accelerate the development of devices for automatic feeding, product conveying, and quality monitoring of printed products. We aspire for 4D printed labels to achieve automated, intelligent, large-scale, and precise production. This will help establish a shelf-life prediction and early warning system of fresh foods, and offer consumers visual information about the quality of fresh foods.

## Methods
### Materials
Fresh kiwi fruits (*Actinidia chinensis* Planch.), green mangoes (*Mangifera indica* Linn.), and persimmons (*Diospyros kaki* Thunb.) were provided by the plantation at Yantai (Shandong, China), Changjiang (Hainan, China), and Baoding (Hebei, China), respectively. The color developer was composed of BA and un-extractable hawthorn polyphenols (A mass ratio of 1:2)[17], and its major chemical compositions were shown in Supplementary Table 1. Soy protein isolate, citrus pectin, garlic essential oil (GEO), konjac glucomannan (KGM), methacrylic anhydride (MA), deuterium oxide, Nile red, ethanol, fluorescein isothiocyanate (FITC), potassium bromide, potassium nitrate, cellulose dialysis membrane (Mw of 8-14 kDa), and additional reagents were sourced from Bailingwei Chemical Technology Co., Ltd. (Shanghai, China). A light-emitting diode with the following parameters was used: wavelength of 445 nm, operating current of 2.0 mA, and power density of 3 W/cm². A polypropylene packaging box (See Supplementary Fig. 5) measuring 181 × 107 × 75 mm had a wall thickness of 0.52 mm and a light transmittance of 80%.

### Classification of quality levels of respiring climacteric fruits
As reported by Teng et al. [10] with slight modifications, kiwi fruits/green mangoes/persimmons of 1 kg were stored in the plastic box containing a sheet of kitchen absorbent paper (Length × width of 200 × 110 mm) and a hygrometer (RH820U, DwyerOmega, America) at 25 °C. Due to the irreversible damage caused by the quality assessment, each type of fruit was divided into 100 test groups for the repeated experiment. Three groups were randomly selected from the 100 test groups to measure relative humidity and carbon dioxide levels in the plastic box daily. Next, the fruit was removed for physicochemical analysis including measurements of hardness, titratable acid content, soluble solid content, weight loss rate, and sensory evaluation. Based on the above chemical indexes, the quality levels of fruits were distinguished using principal component analysis (PCA).

Relative humidity: The value of relative humidity in the plastic box was displayed by the hygrometer[10] (WS101, Kehui Instrument Factory, Tianjin, China).

Carbon dioxide content: An infrared gas analyzer (IR400, Yokogawa Co., Ltd., Yokohama, Japan) was used[32]. Base on the data of

carbon dioxide content, the respiration rate of fruits was calculated as mass of carbon dioxide produced by 1 kg fruits per unit time.

Relationship between carbon dioxide content and pH value: A small glass bottle containing 10 mL pure water was placed in a wide-mouthed bottle. Different volume fractions of premixed gases (carbon dioxide and nitrogen) were introduced into this wide-mouthed bottle at 25 °C. After the gas composition stabilized for 3 min, the pH value of the carbonated solution in the small glass bottle was measured. A functional relationship between the carbon dioxide content and the pH value was established using Origin 2022 software (OriginLab Corporation, Massachusetts, USA). After that, 5 points were randomly selected to measure the actual pH value for verifying the accuracy of the equation. Based on this, the pH range altered by the carbon dioxide produced by fruits was calculated. Subsequently, solutions within this pH range were prepared to test the colorimetric ability of the color developer. The colorimetric ability was represented by RGB (Red, green, and blue) images generated using Adobe Photoshop 2024 software (Adobe Systems, California, USA)[33]. These RGB images displayed the color variations of the color developer at different pH values, compared with pH = 7.0.

Hardness: A texture analyzer (TA. XTPlus, Stable Micro Systems, Surrey, UK) with a P/2 probe was used. Speed before/after test, test speed, a trigger power, and test distance were set at 5.00 mm/s, 1 mm/s, 50 N, and 30 mm, respectively[34].

Titratable acid content: The mixture including 20 g fruit homogenate and 250 mL deionized water was filtered. Then, the acid-base titration method was performed[35].

Soluble solid content: A hand portable refractometer (Pocket Pa-1, ATAGO, Tokyo, Japan) was used to determine the soluble solid content[36]. The sample solution was deposited onto the prism surface, followed by the closure of the cover plate. The instrument was then directed toward a light source for observation until a sharp and stable light-dark boundary appeared in the viewfinder. The value displayed at this point represented the soluble solids content.

Weight loss rate: The weight loss rate was calculated based on the mass difference between the initial weight and the weight after a specified storage period[37].

Sensory evaluation: A trained sensory panel of ten members (Aged between 20 and 38, four males and six females) evaluated the fruits for various sensory parameters namely overall appearance and taste. Recognition and threshold test as well as routine Hedonic tests were performed in the laboratory. The sequence of sample presentation was randomized using the Williams Latin square design, and there was a 5 min break between the samples to cleanse the palate with water and soda biscuits[37]. Panelists were asked to evaluate each sample based on standard 5-point Hedonic scale. The evaluation criteria were provided in Supplementary Table 2[38]. The sensory evaluation was approved by the School of Food Science and Technology at Jiangnan University, confirming that the study complies with local ethical standards, and informed consent was obtained from all participants.

### Preparation of printing substate and antibacterial agent
**Printing substate (MAKGM).** Following the method reported by Teng et al. [18], the mixture of 1% (w/v) KGM and 3% (v/v) MA was prepared at pH = 7.0 and stirred at 300 r/min in a 40 °C water bath for 8 h, after which a ten-fold volume of acetone was added. The precipitate was purified through three ethanol washes, a 5-day dialysis process, and then freeze-drying for storage[39]. The grafting degree was determined to be 15%.

The confirmation of successful grafting was obtained from ¹H-nuclear magnetic resonance spectra (¹H NMR) and attenuated total reflectance-Fourier transform infrared spectra (ATR-FTIR). For ¹H NMR spectra[24], 30 mg sample (Including KGM, the physical mixture of MA and KGM, and MAKGM) was dissolved in 1.2 mL deuterium oxide, transferred to a 5 mm NMR tube, and incubated at 40 °C for 24 h. ¹H

NMR spectra were acquired on a 600 MHz spectrometer (AVANCE NEO, Bruker Co., Switzerland, Germany). For ATR-FTIR spectra[6], 20 mL 1% (w/v) sample solution was poured into round Petri dishes of a 90 mm diameter. These dishes were then placed in an oven at 40 °C for 12 h to produce films. ATR-FTIR spectra of these films were acquired using a Nicolet IS10 spectrometer (Nicolet Co., Madison, America) with 32 scans and a resolution of 4 cm$^{-1}$ across the 4000-800 cm$^{-1}$ range.

**Antibacterial agent (GEO emulsion).** The method was performed as reported by Teng et al.[18] with minor modifications. The emulsifier was synthesized by subjecting a 1:1 (w/w) mixture of soy protein isolate and pectin to 8 cycles of a two-step process. Each cycle consisted of hydration to 75% moisture in a saturated potassium bromide solution at 25 °C for 11 h, followed by microwave vacuum drying at 1.5 W/g for 10 min. The emulsion was then prepared by homogenizing 8 g GEO and 2 g emulsifier in 90 mL water using high-shear (10,000 r/min, 10 min) and high-pressure (40 MPa, 5 passes) steps. The final product was collected by centrifugation (300 × g, 10 min) and stored at 4 °C. The encapsulation efficiency of GEO was 83%.

## Preparation of cast and printed labels

The preparation flowchart for cast and printed labels is shown in Supplementary Fig. 5. The final moisture content of both cast and printed labels was 13% (w/w).

**4D printed labels with different formulations.** A dual nozzle printer (FoodBot D, Shiyin Technology Co., Ltd., Hangzhou, China) was used as the printing tool with a printing nozzle diameter of 0.4 mm and a printing speed of 15 mm/s. As shown in Supplementary Fig. 5, the printed labels were cuboid, featuring the upper and lower structures. The printing ink of the upper structure contained 3% (w/v) MAKGM, 0.2% (w/v) color developer, and 0.08% (w/v) riboflavin. Compared with the upper structure ink, the composition of the lower structure ink did not include the color developer, but contained different mass ratios of GEO emulsion (0%, 6%, 8%, 10%, 12%, and 14%, w/w). At this stage, the upper structure was designed with a single layer, a 30% infill ratio, and a 90° filament interaction angle relative to the lower structure. The lower structure was designed as a single layer with a 100% infill ratio. The printed labels were irradiated using LED light for 5 min and then placed in a fume hood for 12 h. These obtained labels were placed in a dark drying dish for later use.

**4D printed labels with different geometric designs.** According to the deformation degree (Evaluated using the method in section Determination of deformation degree) of printed labels, the optimal addition of GEO emulsion was determined. Subsequently, the influence of different geometric designs of the upper structure on the deformation degree was studied. Supplementary Fig. 7 displays the printed models. These printed models were produced by Rhinoceros 5.0 (Robert McNeel & Associates, Washington, USA), and the STL files were output. The STL files were sliced using Repetier-Host software (Hot-World GmbH & Co. KG, Würzburg, Germany). Then, G-codes were generated to guide the printing path of the 3D printer. The differences in the printing models were reflected in the layer number of the upper structure, the filament intersection angle between the upper and lower structures, and the infill ratio. Each parameter was varied individually whilst keeping the other parameters constant.

(1) The layer number of the upper structure was set to different values of 1, 2, 3, and 4, maintaining an infill ratio of 30% and a filament interaction angle of 90°. (2) The filament interaction angle was set to different values of 0°, 30°, 60°, and 90° with the upper layer number of 1 and an infill ratio of 30%. (3) The infill ratio of the upper structure was set to different values of 30%, 50%, 70%, and 90%, while keeping the upper layer number of 1 and a filament interaction angle of 90°.

**3D printed labels.** Since mixing the color developer with GEO emulsion altered the initial color and monitoring sensitivity, 3D printed labels prepared in this study also featured the upper and lower structures. However, unlike 4D printed labels, the upper layer of 3D printed labels featured an infill ratio of 100%. Preliminary experiments showed that a 100% infill ratio in the upper layer was required to prevent shape deformation in the printed labels. All other preparation steps were consistent with those used for 4D printed labels.

**Cast labels.** Consistent with the preparation of printed labels, cast labels were also composed of the upper and lower structures. MAKGM containing GEO emulsion was used for the lower structure and poured into a petri dish, then dried at 25 °C for 5 h. For the upper structure, MAKGM containing the color developer was poured onto the pre-prepared film[11]. The assembled labels were irradiated with LED light for 5 min and subsequently placed in a fume hood for 12 h. The prepared labels were stored in a light-proof desiccator for subsequent applications.

## Determination of deformation degree

Each label was placed in a drying dish containing a saturated potassium nitrate solution (Relative humidity of 93%) at 25 °C. Each label was tested once. Considering factors such as experimental consumption, parallel test groups, and a 7-day test period, 21 test groups were established for one sample in a replicated experiment. During the testing phase, labels were randomly removed from three drying dishes at 24-hour intervals. Each label was evenly divided into five segments based on its length, and each segment was documented through photography. The curvature of the cross-section was calculated using Image-Pro Plus 6.0 software (Media Cybernetics, Silver Spring, USA)[10]. The degree of deformation was expressed as the average curvature.

## Determination of indexes affecting the deformation degree

**Printing fidelity.** The printing fidelity of the inks used for the upper and lower structures was tested, respectively. The printing parameters were set as follows: a hollow cylinder pattern with dimensions of 10 mm in diameter and 15 mm in height, a printing nozzle diameter of 0.4 mm, and a printing speed of 15 mm/s. After printing, the printed objects were treated using LED irradiation for 5 min, and photographed by a digital camera (OnePlus 9 R, Shenzhen oneplus Technology Company Limited, Shenzhen, China). The printing fidelity was measured using Image-Pro Plus 6.0 software (Media Cybernetics, Silver Spring, USA), and it was calculated using the following Eq. (1)[25]:

$$\text{Printing fidelity(\%)} = \left(1 - \frac{|\text{bottom area of printed object} - \text{bottom area of model}| \times \text{height of printed object}}{\text{bottom area of model} \times \text{height of model}}\right) \times 100$$

(1)

**Surface hydrophobicity.** The materials forming the upper and lower structures were printed into films, respectively. The diameter of a printing nozzle was 0.4 mm, and the infill ratio was 100%. The surface hydrophobicity of these films was quantified by measuring the contact angle using a video contact angle measuring equipment (OCA15EC, Dataphy Instruments Co., Ltd., Nürtingen, Germany)[40]. MilliQ water (10 µL) dropped onto the surface of the film (Length × width of 3 × 3 cm), and the result was captured using a high-speed camera. The contact angle measurements were taken at 5 different points on each film, with a total of 5 films used as replicates.

**Water content.** The printed samples (Length × width of 3 × 3 cm) were stored in a drying dish at 93% relative humidity and 25 °C for 7 days. Samples were taken every 24 h to measure their moisture content. The water content was determined using the weight difference method[41].

Briefly, the tested sample was weighed to get its initial weight ($W_0$), and dried at 105 °C after water absorption to reach a constant weight ($W_1$). Equation (2) for calculating moisture content was as follows:

$$\text{Moisture content}(\%) = \left(\frac{W_0 - W_1}{W_0}\right) \times 100 \qquad (2)$$

**Microstructure.** A field emission scanning electron microscopy (FESEM, Su8100, Hitachi High-Tech Group, Tokyo, Japan) and a confocal laser scanning microscope (CLSM, Axio Vert A1, Carl Zeiss AG, Oberkochen, Germany) were used to characterize the internal structure of the labels. For FESEM images[42], the labels before and after deformation were freeze-dried, and then fractured in liquid nitrogen. Prior to observation, samples were mounted on metal grids and coated with gold under vacuum. Microstructure of the lower structure of the tested labels was observed using FESEM with an accelerating voltage of 3 kV[43,44].

To observe the network structure and GEO distribution before light exposure, the materials used for the upper and lower structures of the labels were poured separately into petri dishes to form a 0.1 mm film. Then, these films (Length × width of 1 × 1 cm) were mounted on glass slides, stained with 50 μL the mixed dye solution (0.1 mg/mL Nile Red and 0.5 mg/mL FITC), and incubated in the dark for 1 h[45]. After applying coverslips, the residual dye was removed by gently wicking from the slide edges. In order to observe the changes in network structure and GEO distribution after light exposure and water absorption, the test labels were sliced into 0.1 mm films using a cryo-sectionaliser (Leica CM3050 S, Leica Biosystems Nussloch GmbH, Nussloch, Germany)[46]. Then, staining was performed as described above. The excited wavenumber was set at 488 nm for Nile Red and 495 nm for FITC.

**Molecular interaction.** Firstly, 4 g printing ink (Used for the upper and lower structures of labels and subjected to LED irradiation) was dissolved in 100 mL deionized water and 8 mol/L urea, called gel A and B, respectively[47]. Then, gel strength (N) was measured using a texture analyzer (TA. XTPlus, Stable Micro Systems, Surrey, UK) with a cylindrical probe of P/0.5, and expressed as the maximum ordinate value of the first peak[10]. The strength of hydrogen bonds was determined as the difference in gel strength between gel A and B. To probe hydrophobic or electrostatic interactions, the urea solution was replaced with 0.1 mol/L sodium dodecyl sulfate or 0.8 mol/L sodium chloride, respectively[47].

**Total porosity.** As described by Teng et al.[10], the weight of samples was determined with an Entris 64-1S electronic balance (Sartorius AG, Göttingen, Germany), and its volume (Length, width, and height) was measured using an Alton M820-25 handheld micrometer (Alton Inspection Technologies, Shanghai, China). The bulk density (g cm$^{-3}$) was calculated according to Eq. (3)[11]. The particle density (g cm$^{-3}$) was determined by liquid displacement using xylene (density = 0.864 g cm$^{-3}$). Prior to this measurement, the samples were cryogenically ground in liquid nitrogen and weighed. The particle density and total porosity were calculated using Eqs. (4) and (5)[11], respectively.

$$\text{Bulk density}(\text{g cm}^{-3}) = \frac{\text{Weight of the dried sample (g)}}{\text{Volume of the dried samples}(\text{cm}^3)} \qquad (3)$$

$$\text{Particle density}(\text{g cm}^{-3}) = \frac{0.864 \times (m_3 - m_1)}{m_2 + (m_3 - m_1) - m_4} \qquad (4)$$

$$\text{Total porosity}(\%) = \left(1 - \frac{\text{bulk density}}{\text{particle density}}\right) \times 100 \qquad (5)$$

Where $m_1$, $m_2$, $m_3$, and $m_4$ were the mass of the empty pycnometer (g), pycnometer + xylene (g), pycnometer + sample (g), and pycnometer + xylene + sample (g), respectively.

## Determination of GEO release
The tested labels were cryogenically ground into small pieces using liquid nitrogen. These pieces were mixed with 95% (v/v) ethyl alcohol, followed by ultrasonic treatment at a specific input power of 4 W/g at 20 °C for 15 min. After centrifugation at 348 ×$g$ for 20 min, the absorbance of the supernatant was measured at 278 nm using an ultraviolet-visible spectrophotometer (UV-1800, Shimadzu, Kyoto, Japan)[28]. The standard curve was established as follows: y (i.e., absorbance) = −3.12156 + 18.12497 x (i.e., GEO concentration). The amount of GEO released was calculated as the difference between the original amount and the measured value. To elucidate the release mechanism, the data were analyzed using the Ritger-Peppas model (Eq. 6)[22]:

$$\text{Ritger} - \text{Peppas} : \frac{M_t}{M_\infty} = k\, t^n + A \qquad (6)$$

where $M_t/M_\infty$, $k$ and $n$ are the release fraction of GEO released at time (%), release rate constant and release exponent, respectively.

## Application in respiring climacteric fruits
Fruit-label pairs (Each containing 1 kg of fruit and a cast/3D printed/4D printed label with dimensions of 4 × 1 cm) were stored in the plastic boxes (Supplementary Fig. 5) at 25 °C, with boxes containing only fruit serving as controls. The measured parameters included sensory evaluations of the fruit's external appearance and state, changes in the color and shape of labels, cumulative GEO release, and total bacterial count. Photographic documentation and quantitative recordings were performed every 24 h. As the measurements were destructive, each box yielded only one data point. For each replication cycle, the experimental groups assigned to a single sample were scaled to triple the total duration of the evaluation period. The lower structure of all labels contained 10% (w/w) GEO emulsion and 0.08% (w/v) riboflavin, whereas the upper structure contained 0.2% (w/v) color developer and 0.08% (w/v) riboflavin. The infill ratio of the lower structure in the printed labels was set to 100%. For the upper structure, the infill ratios of the 3D printed and 4D printed labels were 100% and 30%, respectively. The filament intersection angle was set to 60°.

## Applications of lightweight deep convolutional neural network (DCNN) models
The prediction performance of various combinations between three label types and four lightweight DCNN models was systematically evaluated for monitoring the freshness of respiring climacteric fruits. The evaluation was conducted using an image dataset containing multiple typical fruit types across 3-4 quality levels. According to the method of Teng et al.[10], the images were standardized by normalizing them using the mean and standard deviation of ImageNet. If a custom dataset exhibited a noticeably different distribution, these values were recomputed. Next, data diversity was enhanced through dynamic data augmentation techniques such as random horizontal flipping, rotation, scaling, cropping, and color jittering. Finally, corrupted files and outlier samples were cleaned, and the dataset was divided into an 8:2 train-test split. The models were implemented in PyCharm Community Edition 2023 (JetBrains, Prague, Czech Republic) and evaluated using prediction accuracy[10].

To detect and correct label noise, a multi-model cross-validation strategy was carried out. The specific process was as follows: The dataset was divided into five mutually exclusive subsets. Each subset was served as a validation set in turn while the remaining subsets were used for training. During the cross-validation process, the predictions of each sample across different models were recorded, and samples with predictions from most models contradicting the original label (e.g., 80% predicted class A, but labeled as class B) were identified. These high-confidence conflicting samples were marked as potential annotation error candidates and were subsequently corrected through

semi-automatic cleaning (Filtered based on model prediction probability thresholds). This approach not only effectively identified label noise but also leveraged the complementary nature of different models to reduce the risk of misjudgment, ultimately enhancing the reliability of the dataset labels.

### Statistical analysis

Unless otherwise specified, all indicators were measured three times and each measurement consisted of three parallel groups. All results were collected using Origin 2022 software (OriginLab Corporation, Massachusetts, USA) and presented as mean $\pm$ standard deviation. Statistically significant ($p < 0.05$) was analyzed by one-way ANOVA and a Duncan's test using IBM SPSS Statistics 28 software (IBM Corporation, New York, USA).

### Reporting summary

Further information on research design is available in the Nature Portfolio Reporting Summary linked to this article.

## Data availability

The source data underlying Figs. 1, 2, and 4–9 and Supplementary Figs. 1–3 are available in the associated source data file. Source data are provided with this paper.

## Code availability

The code is provided in the Supplementary Code package, which contains a README file, code for generating the training and test datasets, neural network model architectures, code for model training, code for model evaluation, and a list of Python dependencies.

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

## Acknowledgements

The authors acknowledge financial supports from National Key R&D Program of China (Contracts No. 2022YFD2100601 and No. 2023YFF1104205).

## Competing interests

The authors declare no competing interests.
