## [Transparent Peer Review file · Nature Communications]

4D printed deformation labels with machine learning for monitoring and preservation of respiring climacteric fruits

Corresponding Author: Professor Min Zhang

Version 0:

Reviewer comments:

Reviewer #4

(Remarks to the Author)

The authors have designed a 4D-printed label that visually reflects the ripeness of climacteric fruits by detecting changes in environmental moisture and carbon dioxide during the ripening process. While the authors have a solid experimental and theoretical foundation in the field of fruit preservation, their discussion of the material aspects remains insufficient. Below are my concerns and suggestions:

1. According to the manuscript, the emulsion consists of garlic essential oil, soybean protein, and pectin. In what structural form do these components exist? What are the interactions between the emulsion and MAKGM? This is crucial for understanding how the emulsion affects the mechanical strength and deformation behavior of the material. It is recommended to include schematic diagrams and experimental characterizations to support this discussion.
2. Lines 444–454: Does the term “color developer” refer to blueberry anthocyanins? Please use precise chemical names for reference and check the entire manuscript for similar issues.
3. Line 454: Ensure that the figures appear in the correct order relative to the main text. The content in Lines 448–453 and Figure S2G can be integrated into Section 3.8, and the statement “The research results are presented in Section 3.8.” should be removed.
4. Section 3.4, Line 474: The fidelity and mechanical strength of the ink are fundamental material properties. The authors should consider whether this section belongs in Section 3.4. Additionally, the effect of the emulsion on the ink’s mechanical strength and crosslinking density should be supported with data, such as rheological measurements of the ink.
5. Figure 2E: The scale bar in the SEM image is unclear. Please avoid directly placing the raw image into the figure. Furthermore, Figure 2 lacks a legend for figure E.
6. Line 483: The authors state that the presence of the emulsion inhibits the crosslinking of MAKGM. This raises the same fundamental question as before: how exactly does the emulsion interact with MAKGM? The manuscript lacks an in-depth explanation of the experimental results and fails to pinpoint the core issue.
7. Lines 486–493: What is the underlying reason for this phenomenon? The initial water contact angle increases with increasing emulsion content, yet the water absorption rate first decreases and then increases. How does the material’s surface interact with water? Is this effect driven by the material composition, the structural properties, or both? Additional experiments and explanations are needed.
8. Lines 506–524: The SEM images indicate that as the emulsion content increases, the pore size of the material increases, the porosity decreases, and the crosslinking density increases. Please provide supporting data on the density, porosity, and mechanical strength of the label to substantiate these observations.

9. The manuscript's logical flow is somewhat disorganized. The characterization of MAKGM in Figure 5 would be more appropriately placed at the beginning of the manuscript. Explaining the characterization and interactions of each component at the outset would help clarify many of the mechanistic issues discussed later. The authors are encouraged to refer to well-structured papers published in Nature Communications for guidance on manuscript organization and figure presentation.

Version 1:

Reviewer comments:

Reviewer #4

(Remarks to the Author)

1. We have also conducted some experiments related to garlic essential oil. This oil has a strong and unpleasant odor—could it affect the flavor of fruits?

2. The figure layout and graphical presentation require substantial refinement. The current version demonstrates a notable disparity when compared to contemporary publications in this cutting-edge field. Please consult Nature Communications articles to guide these improvements.

LIST OF OUR RESPONSES TO REVIEWERS' COMMENTS

Manuscript Number: Nature Communications manuscript NCOMMS-25-20212-T

Manuscript title: 4D printed deformation labels: Application in visual monitoring and quality preservation of respiring climacteric fruits

Thanks very much for your email and comments. All of the comments are valuable and very helpful in revising and improving our manuscript. We have studied the comments carefully and have revised the manuscript accordingly. Revised portions are marked in red in the revised manuscript. The following section responses summarizes the main revisions made in the manuscript.

Reviewers' comments:

The authors have designed a 4D-printed label that visually reflects the ripeness of climacteric fruits by detecting changes in environmental moisture and carbon dioxide during the ripening process. While the authors have a solid experimental and theoretical foundation in the field of fruit preservation, their discussion of the material aspects remains insufficient. Below are my concerns and suggestions:

1. According to the manuscript, the emulsion consists of garlic essential oil, soybean protein, and pectin. In what structural form do these components exist? What are the interactions between the emulsion and MAKGM? This is crucial for understanding how the emulsion affects the mechanical strength and deformation behavior of the material. It is recommended to include schematic diagrams and experimental characterizations to support this discussion.

6. Line 483: The authors state that the presence of the emulsion inhibits the crosslinking of MAKGM. This raises the same fundamental question as before: how exactly does the emulsion interact with MAKGM? The manuscript lacks an in-depth explanation of the experimental results and fails to pinpoint the core issue.

Response: Thank you for your affirmative comment. Given the similarity between these two questions, we have consolidated our responses to address them jointly. In this experiment, during the preparation of printing ink, the essential oil emulsion was first added to the aqueous solution, followed by the addition of konjac glucomannan. Therefore, with the aid of the schematic diagram shown in Fig. 3A, we illustrated the existing states among the components. Specifically, the soy protein isolate-pectin conjugate encapsulated the essential oil, and the entire structure was enveloped by a water film and constrained by konjac glucomannan. According to the CLSM images, when konjac glucomannan aggregated upon illumination, the essential oil emulsion appeared in the pores of the formed network, indicating that the interaction between the emulsion and konjac glucomannan was weak. Additionally, as shown in Fig. 3E and F, the porosity increased with the addition of the essential oil emulsion, and Fig. 3G demonstrated that the overall intermolecular forces weakened with the increase of the essential oil emulsion (especially hydrogen bonds). The modified konjac glucomannan required photo-induced crosslinking of its grafted functional groups to form a high-strength network structure. The observed reduction in intermolecular forces caused by the emulsion incorporation suggested that the konjac glucomannan-emulsion interactions were substantially weaker than the intrinsic molecular interactions within the konjac glucomannan matrix. Based on

the data, we thought that the addition of the essential oil emulsion weakened the gel network formed by pure konjac glucomannan, and the added essential oil emulsion was primarily held within the konjac glucomannan by mechanical or very weak intermolecular forces. Please see lines 460–483.

2.Lines 444–454: Does the term “color developer” refer to blueberry anthocyanins? Please use precise chemical names for reference and check the entire manuscript for similar issues.

Response: Thank you for your affirmative comment. The color developer was composed of blueberry anthocyanins and un-extractable hawthorn polyphenols (A mass ratio of 1:2). The major chemical compositions of blueberry anthocyanins and un-extractable hawthorn polyphenols were shown in Table S1. In the resubmitted manuscript, we explained the composition of the color developer. Please see lines 146-147. For simplicity, we retained the term “color developer.”

3.Line 454: Ensure that the figures appear in the correct order relative to the main text. The content in Lines 448–453 and Figure S2G can be integrated into Section 3.8, and the statement “The research results are presented in Section 3.8.” should be removed.

Response: Thank you for your affirmative comment. The article’s framework was adjusted according to your suggestions.

4.Section 3.4, Line 474: The fidelity and mechanical strength of the ink are fundamental material properties. The authors should consider whether this section belongs in Section 3.4. Additionally, the effect of the emulsion on the ink’s mechanical strength and crosslinking density should be supported with data, such as rheological measurements of the ink.

Response: Thank you for your affirmative comment. Structural design was one of the decisive factors for achieving the functional performance of printed labels, which required high printing precision. The amount of essential oil added affected printing accuracy. Therefore, we considered printing precision as one of the factors influencing label functionality. Moreover, the storage modulus, loss modulus, and $\tan\delta$ of the printing inks containing different essential oil emulsions before and after light exposure were added. Relevant analyses can be found in the Lines 511~524 and Fig. S2.

5.Figure 2E: The scale bar in the SEM image is unclear. Please avoid directly placing the raw image into the figure. Furthermore, Figure 2 lacks a legend for figure E.

Response: Thank you for your affirmative comment. The image resolution was optimized for clarity, and a scale bar were added to Fig. 2E.

7.Lines 486–493: What is the underlying reason for this phenomenon? The initial water contact angle increases with increasing emulsion content, yet the water absorption rate first decreases and then increases. How does the material’s surface interact with water? Is this effect driven by the material composition, the structural properties, or both? Additional experiments and explanations are needed.

Response: Thank you for your affirmative comment. The surface hydrophobicity of printed products and the density of their formed network structure significantly affected water contact angles at different time intervals. When the essential oil content in the emulsion increased ($\leq 10\%$, w/w), the essential oil components formed an effective waterproof barrier that prevented water infiltration.

Concurrently, the ink exhibited a relatively dense network structure (as shown in the SEM image in Fig. 4E), further hindering water penetration. However, samples containing excessive emulsion initially demonstrated larger contact angles due to their stronger hydrophobicity. Over time, water gradually penetrated into the printed product. Since excessive essential oil addition disrupted the network structure, creating larger pores (as confirmed by the SEM results in Fig. 4E), the water resistance of these samples deteriorated in later observation stages, ultimately resulting in reduced contact angles. Please see lines 539~551.

8.Lines 506–524: The SEM images indicate that as the emulsion content increases, the pore size of the material increases, the porosity decreases, and the crosslinking density increases. Please provide supporting data on the density, porosity, and mechanical strength of the label to substantiate these observations.

Response: Thank you for your affirmative comment. SEM images revealed that as the emulsion content increased, the material's pore size expanded and the crosslinking density decreased. Data on the total porosity of the labels before and after deformation were added in the resubmitted manuscript. Please see Fig. 4F. The mechanical strength was demonstrated by the rheological results shown in Fig. S1.

9.The manuscript's logical flow is somewhat disorganized. The characterization of MAKGM in Figure 5 would be more appropriately placed at the beginning of the manuscript. Explaining the characterization and interactions of each component at the outset would help clarify many of the mechanistic issues discussed later. The authors are encouraged to refer to well-structured papers published in Nature Communications for guidance on manuscript organization and figure presentation.

Response: Thank you for your affirmative comment. In the resubmitted manuscript, we have adjusted the logical framework according to your suggestions.

We have corrected the manuscript according to the reviewers' comments and believe that the revisions made have significantly improved its quality.

We look forward to your positive response and information about our revised manuscript.

Thank you once again!

With Best regards,

Min Zhang, Professor

LIST OF OUR RESPONSES TO REVIEWERS' COMMENTS

Manuscript Number: Nature Communications manuscript NCOMMS-25-20212A

Manuscript title: 4D printed deformation labels: Application in visual monitoring and quality preservation of respiring climacteric fruits

Thanks very much for your email and comments. All of the comments are valuable and very helpful in revising and improving our manuscript. We have studied the comments carefully and have revised the manuscript accordingly. Revised portions are marked in red in the revised manuscript. The following section responses summarizes the main revisions made in the manuscript.

Reviewers' comments:

1. We have also conducted some experiments related to garlic essential oil. This oil has a strong and unpleasant odor—could it affect the flavor of fruits?

Response: Thank you for your suggestion. Garlic essential oil, characterized by inherent hydrophobicity and a strong pungent odor, can be effectively encapsulated within emulsion film matrices via emulsifiers, with concurrent reduction of its characteristic aroma intensity (de Souza et al., 2020; Katata-Seru et al., 2017; Song et al., 2018). According to the published reports, garlic essential oil has now been incorporated into films to extend the shelf life of some fruits and vegetables, such as cherry tomatoes (Li et al., 2024), bananas (Khaliq et al., 2019, Orsuwan & Sothornvit, 2018), grapes (Xie et al., 2022), and strawberries (Dong & Wang, 2017). Considering the combined effects of odor control and antibacterial efficacy, the garlic essential oil content in these films was limited to $\leq 2\%$ (w/w) of the total film-forming solution mass. The present study employed a concentration of 0.8% (w/w), resulting in merely a trace garlic aroma detectable on labels (only upon deliberate olfactory inspection) without compromising the fruits' characteristic fragrance. Thank you again for your valuable suggestions.

References:

1. de Souza, A. G., Agostinho dos Santos, N. M., da Silva Torin, R. F., & Rosa, D. d. S. (2020). Synergic antimicrobial properties of Carvacrol essential oil and montmorillonite in biodegradable starch films. *International Journal of Biological Macromolecules*, 164, 1737-1747.
2. Dong, F., & Wang, X. (2017). Effects of carboxymethyl cellulose incorporated with garlic essential oil composite coatings for improving quality of strawberries. *International Journal of Biological Macromolecules*. 104, 821-826.
3. Katata-Seru, L., Lebepe, T. C., Aremu, O. S., & Bahadur, I. (2017). Application of Taguchi method to optimize garlic essential oil nanoemulsions. *Journal of Molecular Liquids*, 244, 279-284.
4. Khaliq, G., Abbas, H. T., Ali, I., & Waseem, M. (2019). Aloe vera gel enriched with garlic essential oil effectively controls anthracnose disease and maintains postharvest quality of banana fruit during storage. *Horticulture, Environment, and Biotechnology*. 60, 659–669.
5. Li, L., Zhao, Z., Wei, S., Xu, K., Xia, J., Wu, Q., . . . Wang, L. (2024). Development and

application of multifunctional films based on modified chitosan/gelatin polyelectrolyte complex for preservation and monitoring. *Food Hydrocolloids*, 147, 109336.

6. Orsuwan, A., & Sothornvit, R. (2018). Active Banana Flour Nanocomposite Films Incorporated with Garlic Essential Oil as Multifunctional Packaging Material for Food Application. *Food and Bioprocess Technology*.11, 1199-1210.
7. Song, X., Zuo, G., & Chen, F. (2018). Effect of essential oil and surfactant on the physical and antimicrobial properties of corn and wheat starch films. *International Journal of Biological Macromolecules*, 107, 1302-1309.
8. Xie, Y., Zhu, J., Liu, H., Lian, H., & Liu, J. (2022). In vitro antifungal activity of essential oils against *Botrytis cinerea* of postharvest grapes. *IOP Conference Series: Earth and Environmental Science*. 1035, 012008.

2. The figure layout and graphical presentation require substantial refinement. The current version demonstrates a notable disparity when compared to contemporary publications in this cutting-edge field. Please consult Nature Communications articles to guide these improvements.

Response: Thank you for your suggestion. In the resubmitted manuscript, the formatting of figures and tables has been revised. Please see lines 1006~1068.

We have corrected the manuscript according to the reviewers' comments and believe that the revisions made have significantly improved its quality.

We look forward to your positive response and information about our revised manuscript.

Thank you once again!

With Best regards,

Min Zhang, Professor